# VISUAL ANALYSIS OF THE BUMPINESS AND RUGGEDNESS OF RESIDUAL NEURAL NETWORK LANDSCAPES

## ABSTRACT

Different neural network architectures result in distinct training and generalization results. For instance, deep residual neural networks are more likely to find better local minima and perform more accurate predictions than deep non-residual neural networks. However, the causes of this phenomenon still require clarification. Some works show that, for convolutional neural networks, the residual connection is beneficial to generating smooth loss landscapes. However, our visual analysis discovers opposite conclusions for MLPs. Specifically, in the XOR, Iris, and Diabetes datasets, residual MLPs tend to produce more rugged loss landscapes with stronger gradients and higher loss values than non-residual MLPs, but they can still converge to low-loss basins of attraction. In the XOR dataset, residual MLPs prefer to generate more attraction basins that are sharper and have lower loss values than non-residual MLPs. Our work advances the knowledge of residual connections.

## 1 INTRODUCTION

Residual connections were first proposed by He et al. (2016) for convolutional neural networks (CNNs). Residual neural networks (ResNets) (He et al. (2016)) effectively resolve the gradient vanishing problem when the network depth increases. The presence of residual connections is a crucial factor for the achievements of deep neural networks. However, the effect of residual connections on deep neural networks requires further investigation. Some works have studied the loss landscapes of residual CNNs. Li et al. (2018) performed empirical studies and found that deep residual CNNs show smoother loss landscapes than deep non-residual CNNs, followed by which Yao et al. (2020) discovered that non-residual CNNs produce more rugged loss landscapes than residual CNNs. In addition, the properties of basins of attraction and local minima in the loss landscapes are debated in many works. Yao et al. (2020) observed that, in deep CNNs, removing batch normalization (BN) layers results in sharp local minima with high loss values and poor generalization ability. Bosman et al. (2023b) discovered that, in MLPs, high-dimensional neural networks have deeper, sharper basins of attraction than lower-dimensional neural networks.

To further advance people's understanding of residual connections, our work concentrates on the effect of residual connections on MLPs, which are challenging to train because of their numerous parameters and show weak performance in visual tasks compared to CNNs. However, recent works in Touvron et al. (2022) (ResMLP) and Kim (2021) have demonstrated that residual MLPs can attain competitive performance in visual tasks. In addition, MLPs play an essential role in non-visual tasks, such as predicting the lifespan of materials and various diseases. Reasonable visualization of the loss landscapes is a common target in many studies, which can enhance the intuitive understanding of the loss landscapes. One of the most interesting works is the loss-gradient cloud introduced by Bosman et al. (2020a), and it is a two-dimensional visualization method for visualizing the loss landscapes, which successfully captures the local minima properties in the loss landscapes and can reflect the curvature information of the loss landscapes. However, there are several shortcomings of the loss-gradient cloud: (i) the loss-gradient cloud classifies the loss landscapes into four curvature types: convex, concave, saddle, and singular, using the Hessian matrix, but it is unable to distinguish the difference in the curvature of the loss landscapes with the same type; (ii) the two-dimensional presentation method can capture some characteristics of the loss landscapes, but the results represented are still not clear enough; and (iii) the loss-gradient cloud considers the loss values, gradient

magnitude, and curvature types of the loss landscapes, but it does not consider the ruggedness of the loss landscapes, which is an essential metric in measuring the network performance.

To address the above problems, our work proposes a three-dimensional visualization method, referred to as the Hessian clouds, for observing the loss landscapes. Our inventive contributions are as follows:

- We derive mathematical expressions of the bumpiness and ruggedness metrics of the loss landscapes, based on which the sharpness (sensitivity) metric can also be obtained.
- Based on the bumpiness and ruggedness metrics, we propose Hessian clouds, a new method for visualizing the loss landscapes, which can display the loss landscapes in three-dimensional spaces.
- We empirically compare the loss landscapes of residual MLPs and non-residual MLPs using the Hessian clouds.

## 2 RELATED WORKS

Numerous works studied the loss landscapes to comprehend neural networks. Bosman et al. investigated the impact of search space boundaries (Bosman et al. (2016)), regularization (Bosman et al. (2018b)), architecture settings (Bosman et al. (2020b)), and different activation functions (Bosman et al. (2023a)) on the loss landscapes. Pimenta et al. (2020) analyzed automated machine learning (AutoML) using fitness landscape analysis (FLA). To investigate neural architecture search problems, Traoré et al. (2021) presented the fitness landscape footprint. Rodrigues et al. (2022) studied convolutional neural network architectures using FLA.

People's intuition about network architectures can be enhanced through the visualization of the loss landscapes, and various metrics have been devised to quantify the structure of the loss landscapes. Kordos & Duch (2004) draw 3-dimensional plots of the loss landscapes using PCA. Li et al. (2018); Bain et al. (2021) introduced a "filter normalization" method and utilized multiple visualization methods to depict the loss landscapes. Bosman et al. (2020a) proposed the loss-gradient cloud to visualize the attraction basins in the loss landscapes. Bosman et al. (2023b) measured the sharpness of the attraction basins and introduced a two-dimensional visualization method for analyzing the basins of attraction. The ruggedness of the loss landscapes was estimated by Kauffman & Levin (1987) based on the average length of the adaptive walk. Entropy was used by Malan & Engelbrecht (2009) to calculate the ruggedness of the loss landscapes. By using the top eigenvalue of the Hessian matrix, Dong et al. (2019) measured the sensitivity of the loss landscapes. Dong et al. (2020) utilized the average Hessian trace as the sensitivity metric. Yao et al. (2020) proposed a new framework called PYHESSIAN, which can efficiently compute Hessian information, and applied it to analyze the effect of residual connections and batch normalization (BN) on CNNs.

FLA requires sampling points in the loss landscapes. Different sampling algorithms yield distinct analytical results. Mersmann et al. (2011) introduced the Latin hypercube as a sampling technique to increase search space coverage. The random walk is a technique for randomly sampling from a given probability distribution (Malan & Engelbrecht (2014); Weinberger (1990)). Pitzer & Affenzeller (2011) discovered that the adaptive walk typically performs better than the random walk, which can avoid getting points with poor fitness values. Malan & Engelbrecht (2013) implemented the Manhattan progressive random walk to efficiently evaluate the loss landscapes. The progressive random walk is a directionally biased variant of the random walk proposed by Malan & Engelbrecht (2014). Bosman et al. (2018a) introduced a gradient-based sampling algorithm called progressive gradient walk (PGW), which can find landscapes with good fitness values.

## 3 METHODOLOGY

### 3.1 BUMPINESS AND RUGGEDNESS METRICS

The notion of bumpiness is also known as roughness or ruggedness (Park et al. (2018)). Park et al. (2018) defined the bumpiness of one-dimensional functions as $B(f(x)) = \int |f''(x)|^2 dx$. However, in our work, we want to quantify the convexity or concavity of the functions. Therefore, we

redefine the bumpiness metric as the degree of convexity or concavity of a function to clearly distinguish it from the ruggedness metric. We define the bumpiness and ruggedness of one-dimensional functions as $B(f(x)) = \int f''(x)dx$ and $R(f(x)) = \int |f''(x)|dx$. For multidimensional functions, we define $B(f(\boldsymbol{x})) = \sum_{i=1}^{n} \int \frac{\partial^2 f}{\partial x_i^2}dx_i$ and $R(f(\boldsymbol{x})) = \sum_{i=1}^{n} \int |\frac{\partial^2 f}{\partial x_i^2}|dx_i$. For bumpiness and ruggedness in an infinitesimal domain of point $\boldsymbol{x}^*$ of a multidimensional function, we have $B(\boldsymbol{x}^*) = \sum_{i=1}^{n} \frac{\partial^2 f}{\partial x_i^2}(\boldsymbol{x}^*)dx_i$ and $R(\boldsymbol{x}^*) = \sum_{i=1}^{n} |\frac{\partial^2 f}{\partial x_i^2}(\boldsymbol{x}^*)|dx_i$. However, this differential expression is unable to identify the magnitude of bumpiness and ruggedness of the loss landscapes surrounding point $\boldsymbol{x}^*$. Therefore, we next use a proof to derive new mathematical expressions of the bumpiness and ruggedness of points in the loss landscapes:

**Assumption 1** *Assume that:*

- *The loss function is second-order differentiable.*

- *The third and subsequent terms in the Taylor expansion of the loss function are very small and can be ignored.*

**Lemma 1** *Given a loss function $L(\boldsymbol{\theta})$ and its corresponding tangent plane $T(\boldsymbol{\theta})$ at point $\boldsymbol{\theta}^*$, we have:*

$$L(\boldsymbol{\theta}^* + \Delta\boldsymbol{\theta}_i) - T(\boldsymbol{\theta}^* + \Delta\boldsymbol{\theta}_i) = \frac{1}{2}||\Delta\boldsymbol{\theta}_i||^2\lambda_i$$

*where $\lambda_i$ is the ith eigenvalue of the Hessian matrix $\boldsymbol{H}(L)(\boldsymbol{\theta}^*)$; $\boldsymbol{H}(L)(\boldsymbol{\theta}^*)$ is the Hessian matrix of $L(\boldsymbol{\theta})$ at point $\boldsymbol{\theta}^*$; the direction of $\Delta\boldsymbol{\theta}_i$ is the same as the direction of the ith eigenvector of $\boldsymbol{H}(L)(\boldsymbol{\theta}^*)$; and $||\Delta\boldsymbol{\theta}_i|| \to 0$.*

**Proof 1** *Given a loss function $L(\boldsymbol{\theta})$ and its corresponding tangent plane $T(\boldsymbol{\theta})$ at point $\boldsymbol{\theta}^*$ based on Assumption 1. $L(\boldsymbol{\theta})$ and $T(\boldsymbol{\theta})$ have the same value and gradient at point $\boldsymbol{\theta}^*$. By using the Taylor expansion, we have:*

$$L(\boldsymbol{\theta}^* + \Delta\boldsymbol{\theta}_i) = L(\boldsymbol{\theta}^*) + \nabla_{\boldsymbol{\theta}}L^T\Delta\boldsymbol{\theta}_i + \frac{1}{2}\Delta\boldsymbol{\theta}_i^T\boldsymbol{H}(L)(\boldsymbol{\theta}^*)\Delta\boldsymbol{\theta}_i$$

$$T(\boldsymbol{\theta}^* + \Delta\boldsymbol{\theta}_i) = T(\boldsymbol{\theta}^*) + \nabla_{\boldsymbol{\theta}}T^T\Delta\boldsymbol{\theta}_i + \frac{1}{2}\Delta\boldsymbol{\theta}_i^T\boldsymbol{H}(T)(\boldsymbol{\theta}^*)\Delta\boldsymbol{\theta}_i$$

*where $\boldsymbol{H}(T)(\boldsymbol{\theta}^*) = 0$ because $T(\boldsymbol{\theta})$ is a plane, and for $i$ from 1 to n, we might as well make the values of $||\Delta\boldsymbol{\theta}_i||$ the same, equal to $\epsilon > 0$.*

*Calculate the difference between the two surfaces in an infinitesimal domain around point $\boldsymbol{\theta}^*$ in the direction of $\Delta\boldsymbol{\theta}_i$:*

$$L(\boldsymbol{\theta}^* + \Delta\boldsymbol{\theta}_i) - T(\boldsymbol{\theta}^* + \Delta\boldsymbol{\theta}_i) = \frac{1}{2}\Delta\boldsymbol{\theta}_i^T\boldsymbol{H}(L)(\boldsymbol{\theta}^*)\Delta\boldsymbol{\theta}_i = \frac{1}{2}\Delta\boldsymbol{\theta}_i^T\boldsymbol{Q}_l\boldsymbol{\Lambda}_l\boldsymbol{Q}_l^T\Delta\boldsymbol{\theta}_i$$

*where the orthogonal matrix $\boldsymbol{Q}_l$ consists of the unitized eigenvectors $\{\boldsymbol{v}_1, \boldsymbol{v}_2, \ldots, \boldsymbol{v}_n\}$ of $\boldsymbol{H}(L)(\boldsymbol{\theta}^*)$ and the diagonal matrix $\boldsymbol{\Lambda}_l$ is composed of the eigenvalues of $\boldsymbol{H}(L)(\boldsymbol{\theta}^*)$. Due to the orthogonality of the real symmetric matrix eigenvectors and $||\boldsymbol{v}_i|| = 1$, we have:*

$$\Delta\boldsymbol{\theta}_i^T\boldsymbol{Q}_l = [0, \ldots, 0, \epsilon, 0, \ldots, 0] = \epsilon\boldsymbol{e}^{(i)}$$

*where $\boldsymbol{e}^{(i)}$ is a standard basis vector $[0, \ldots, 0, 1, 0, \ldots, 0]$ with a 1 at position $i$, and we have:*

$$L(\boldsymbol{\theta}^* + \Delta\boldsymbol{\theta}_i) - T(\boldsymbol{\theta}^* + \Delta\boldsymbol{\theta}_i) = \frac{1}{2}\epsilon^2\boldsymbol{e}^{(i)}\boldsymbol{\Lambda}_l\boldsymbol{e}^{(i)T} = \frac{1}{2}\epsilon^2\lambda_i$$

*where $\lambda_i$ is the ith eigenvalue of $\boldsymbol{H}(L)(\boldsymbol{\theta}^*)$. The geometric meaning of $L(\boldsymbol{\theta}^* + \Delta\boldsymbol{\theta}_i) - T(\boldsymbol{\theta}^* + \Delta\boldsymbol{\theta}_i)$ is the difference between a surface and its corresponding tangent plane in an infinitesimal domain at the tangent point $\boldsymbol{\theta}^*$ in one direction. This gap is a second-order infinitesimal number with respect to $\epsilon$.*

**Definition 1** *In Lemma 1, although $||\Delta\boldsymbol{\theta}_i||$ is infinitesimal, $\lambda_i$ can reflect the magnitude of $L(\boldsymbol{\theta}^* + \Delta\boldsymbol{\theta}_i) - T(\boldsymbol{\theta}^* + \Delta\boldsymbol{\theta}_i)$ to some extent. Therefore, we define the bumpiness in a single direction of point $\boldsymbol{\theta}^*$, i.e., the directional bumpiness as:*

$$B(\boldsymbol{\theta}^*)_i = \lim_{\Delta\boldsymbol{\theta}_i \to 0} \frac{L(\boldsymbol{\theta}^* + \Delta\boldsymbol{\theta}_i) - T(\boldsymbol{\theta}^* + \Delta\boldsymbol{\theta}_i)}{||\Delta\boldsymbol{\theta}_i||^2} = \frac{1}{2}\lambda_i$$

With Lemma 1 and Definition 1, we can derive the following two definitions:

**Definition 2** *The ruggedness of point $\boldsymbol{\theta}^*$ is defined as the average of the sum of the absolute values of the directional bumpiness in all eigenvectors' directions:*

$$R(\boldsymbol{\theta}^*) = \frac{1}{n}\sum_{i=1}^{n}|B(\boldsymbol{\theta}^*)_i| \propto \frac{1}{n}\sum_{i=1}^{n}|\lambda_i|$$

*where $R(\boldsymbol{\theta}^*) = 0$ indicates that the landscape around point $\boldsymbol{\theta}^*$ is strictly smooth.*

**Definition 3** *The bumpiness of point $\boldsymbol{\theta}^*$ is defined as the average of the sum of the directional bumpiness in all eigenvectors' directions, divided by the ruggedness of point $\boldsymbol{\theta}^*$:*

$$B(\boldsymbol{\theta}^*) = \frac{\frac{1}{n}\sum_{i=1}^{n}B(\boldsymbol{\theta}^*)_i}{R(\boldsymbol{\theta}^*)} \propto \frac{Tr(\boldsymbol{H}(L)(\boldsymbol{\theta}^*))}{\sum_{i=1}^{n}|\lambda_i|}, \ if \ R(\boldsymbol{\theta}^*) \neq 0$$

*where $B(\boldsymbol{\theta}^*) \in [-1, 1]$, $B(\boldsymbol{\theta}^*) = 1$ indicates convex (not strict) landscapes, $B(\boldsymbol{\theta}^*) = -1$ indicates concave (not strict) landscapes, and $B(\boldsymbol{\theta}^*) = 0$ represents saddle landscapes.*

Dong et al. (2020) defined the sensitivity (sharpness) metric of convex landscapes, similar to the bumpiness and ruggedness metrics. When the loss landscape is convex, we can derive the similar sharpness metric like Dong et al. (2020) from bumpiness and ruggedness: it's easy to find that when $B(\boldsymbol{\theta}^*) = 1$, i.e., the loss landscape around point $\boldsymbol{\theta}^*$ is convex, the bigger the $R(\boldsymbol{\theta}^*)$, the shaper the loss landscape.

### 3.2 HESSIAN CLOUDS

The loss-gradient cloud, proposed by Bosman et al. (2020a), is a two-dimensional visualization technique for the loss landscapes, which reflects the structure of loss landscapes using the loss values, gradient magnitude ($L^2$ norm of the gradient), and Hessian information of the neural networks. However, the loss-gradient cloud is difficult to work in the situations shown in the Table 1 of Appendix A.2, where almost all the loss landscapes found are classified as singular (with eigenvalues of the Hessian matrix equal to 0). The bumpiness metric can solve this problem (the Figure 1 of Appendix A.1). Further, as discussed in Introduction 1, the loss-gradient cloud cannot show the loss landscapes clearly enough, and it does not take into account the ruggedness metric, which has a great influence on the network performance. To deal with this problem, we visualize the loss landscapes at a finer granularity in three-dimensional spaces, based on the above definitions of bumpiness and ruggedness. We refer to this visualization method as Hessian clouds, as shown in Figure 1. The color of the scatters in Hessian clouds is determined by their bumpiness. The three axes of the three-dimensional spaces are loss values, gradient magnitude, and ruggedness, respectively.

## 4 EXPERIMENTAL SETTING

### 4.1 SAMPLING SETUP

We apply the progressive gradient walk (PGW) proposed by Bosman et al. (2018a) as the sampling algorithm. The random initialization of neural networks' weights is in the range of $[-1, 1]$ with a uniform distribution. There are $10 \times d$ walks, where $d$ equals the output dimension of the neural networks, and each walk consists of 1000 steps with a maximal step size of 0.01. We set $10 \times d$ different random seeds to initialize the neural networks' weights for each walk.

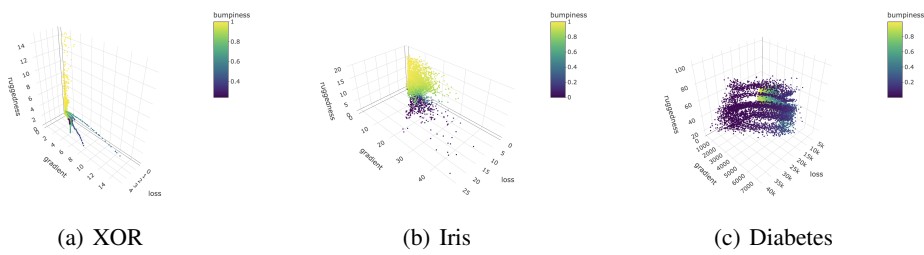

(a) XOR          (b) Iris          (c) Diabetes

Figure 1: various structures in Hessian clouds

## 4.2 NEURAL NETWORK ARCHITECTURE

Our work studies residual (Figure 2(a)) and non-residual (Figure 2(b)) fully connected neural networks (MLPs). The architecture of the non-residual MLPs used in our experiments consists of an input layer, multiple hidden layers, and an output layer. All hidden layers have the same dimension equal to the input layer's dimension. The number of hidden layers is 2, 3, or 4, representing three different network depths. The ReLU activation function follows the input layer and each hidden layer. For residual MLPs, the input feature of the hidden layer is added to the corresponding output feature that the hidden layer has processed to form residual connections.

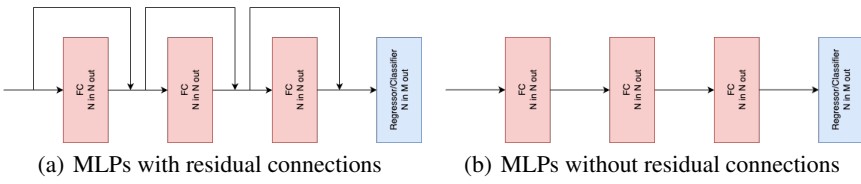

(a) MLPs with residual connections          (b) MLPs without residual connections

Figure 2: residual MLPs and non-residual MLPs

## 4.3 DATASETS

We use three datasets for our experiments: the XOR, Iris (Fisher (1936)), and Diabetes (Efron et al. (2004)) datasets. The dimension of the input features of the XOR dataset is 2, and the output features' dimension is 1. The neural network is evaluated using the MSE error function. For the Iris dataset, the neural network has an input dimension of 4 and an output dimension of 3, optimized by the cross-entropy loss function. The input features' dimension of the Diabetes dataset is 10, and the Diabetes dataset is a regression problem. Therefore, the MSE loss function is used for optimization.

## 5 EXPERIMENT RESULTS

Appendix A.3 illustrates the Hessian clouds of the XOR, Iris, and Diabetes datasets from diverse perspectives. The Table 1 of Appendix A.2 displays the statistics of different types of points sampled in the loss landscapes. It is easy to observe that singular points dominate the loss landscapes of the MLPs used in our experiments using the ReLU activation function.

### 5.1 XOR

Figure 3 represents the entire landscapes found in all walks, revealing that residual MLPs produce more rugged landscapes with stronger gradients than non-residual MLPs, whose large gradients may be advantageous to gradient-based optimization (Solla et al. (1988)).

It is evident from Appendix A.4 and Figure 4 that the loss landscapes contain numerous V-shaped oscillation areas. Each V-shaped oscillation area is a basin of attraction and has a stationary attractor at the vertex of V. Nearly all of the basins are singular, but the corresponding bumpiness

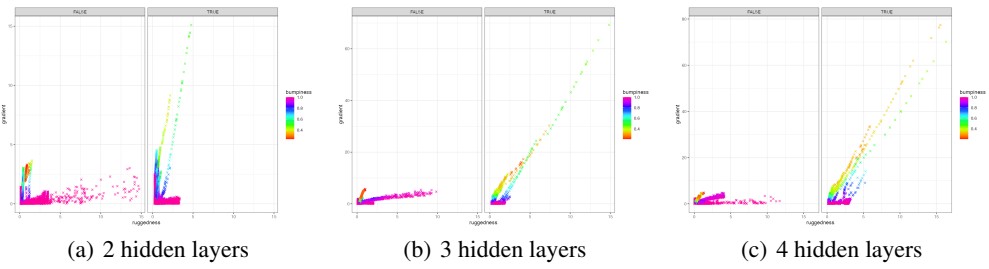

(a) 2 hidden layers      (b) 3 hidden layers      (c) 4 hidden layers

Figure 3: XOR, entire landscapes

is nearly equal to one, indicating that these basins are almost convex. With NN depth deepening, residual MLPs have shaper basins than non-residual MLPs, which will likely benefit gradient-based optimization. In deep NNs, residual MLPs have more basins than non-residual MLPs.

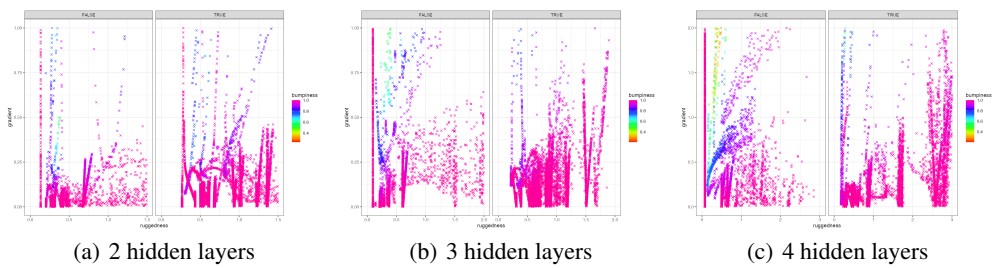

(a) 2 hidden layers      (b) 3 hidden layers      (c) 4 hidden layers

Figure 4: XOR, V-shaped oscillation areas

Figure 5 shows that along the walking paths, the loss landscapes become increasingly convex (from non-convex to flat convex to sharp convex), and the more rugged the loss landscapes of the weight initialization areas are, the easier it is to converge to sharper basins. Almost no connection exists between the basins. In deep NNs, the loss landscapes of residual MLPs contain more low-loss basins than those of non-residual MLPs, indicating that it is simpler for MLPs with residual connections to converge to low-loss landscapes.

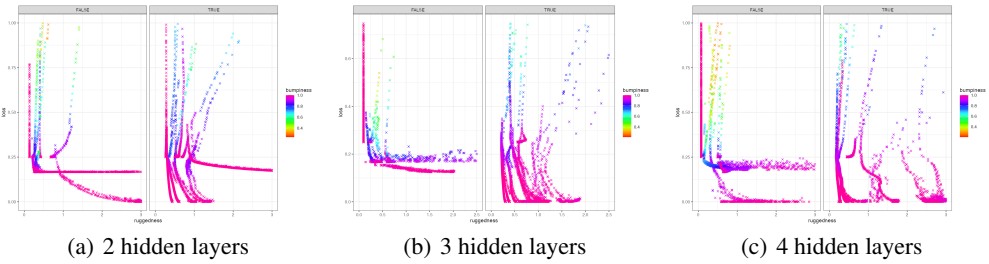

(a) 2 hidden layers      (b) 3 hidden layers      (c) 4 hidden layers

Figure 5: XOR, rugged landscapes tends to fall into sharp basins

Figure 6 demonstrates that non-residual MLPs perform several walks that cannot converge to basins in 1000 steps but instead continue to walk towards sharper convex landscapes, indicating that other sharp basins may be discovered after 1000 steps.

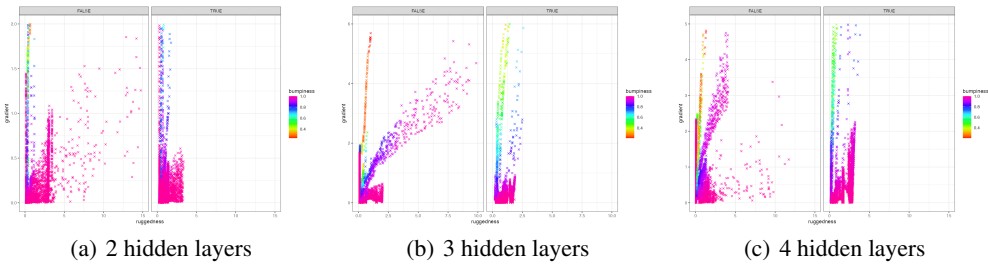

Figure 6: XOR, undiscovered basins in the loss landscapes of non-residual MLPs

As depicted in Figure 7, the walks reach the attractors at the V's vertex and then convulse in the basins. In addition, the paths in deep networks tend to converge first to non-stationary attractors and then to stationary attractors (i.e., the vertex of V).

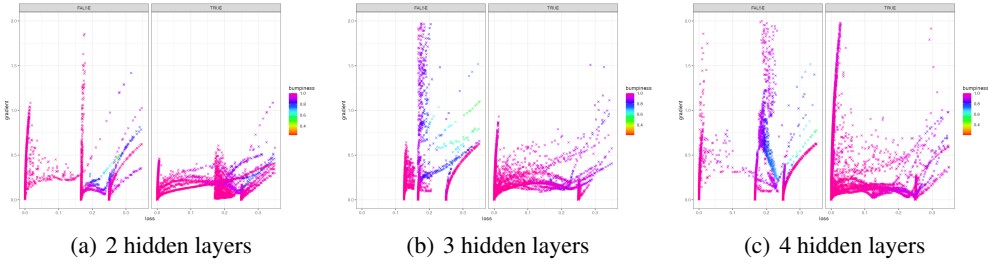

Figure 7: XOR, non-stationary attractors

## 5.2 IRIS

Figure 8 and Figure 9 demonstrate that residual MLPs have landscapes that are more rugged, steeper, and have greater loss values than non-residual MLPs, and both the residual and non-residual MLPs converge to a basin of attraction. The basin has a stationary attractor and nearly convex landscapes. The flat convex landscapes (ruggedness close to 0) close to the stationary attractor may cause subsequent walks to disperse. Observation of the diffusion behavior reveals that walks diffuse into various convex landscapes with large and small sharpness and gradients, and with similar loss values close to 0. The sharper and steeper the basin of the loss landscapes of residual MLPs compared to non-residual MLPs indicates that residual MLPs are less challenging to be optimized.

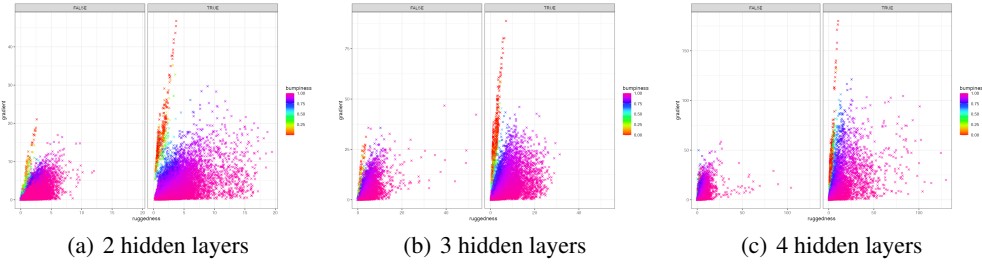

Figure 8: Iris, basins of attraction

Figure 9 depicts a transition from non-convex to convex curvature surrounding the stationary attractors, and the landscapes of the weight initialization areas are extremely saddle and smooth (bumpiness is near 0 and ruggedness is small, as shown in Figure 8 and the Figure 3 of Appendix A.3).

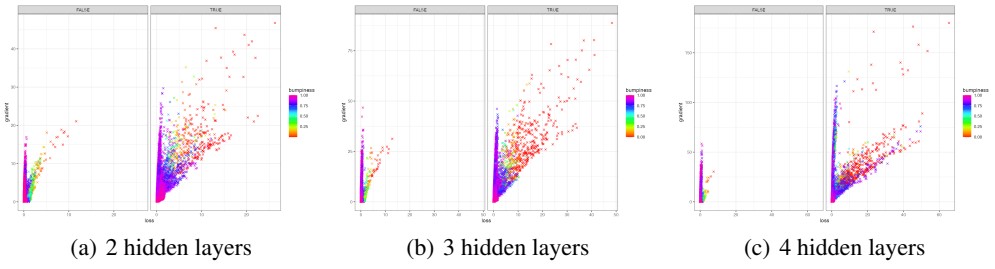

(a) 2 hidden layers      (b) 3 hidden layers      (c) 4 hidden layers

Figure 9: Iris, curvature change

Appendix A.5 and Figure 10 represent that non-residual MLPs also converge to a low-gradient basin with higher loss values. The attraction basin has convex landscapes with low gradients around the attractor and non-strictly convex landscapes with higher gradients further away from the attractor. Walks oscillate back and forth from convex to non-convex landscapes in the basin, and the oscillation of the gradient in the basin has a tiny range.

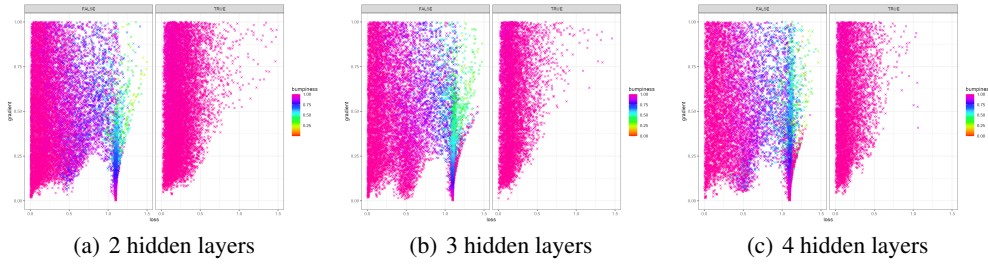

(a) 2 hidden layers      (b) 3 hidden layers      (c) 4 hidden layers

Figure 10: Iris, another attractor of non-residual MLPs

## 5.3 DIABETES

Figure 11 and the Figure 4 of Appendix A.3 depict paths that converge along arches into attraction basins. On the arches, the landscapes transition from low gradients and high loss values to high gradients and medium loss values, and then to low gradients and low loss values. All arches transition from saddle to convex curvature, and the convex landscapes correspond to the basins with stationary attractors. Figure 12 demonstrates that all arches eventually converge to their respective low-loss basins, and these basins have similar loss values. Along the paths, the ruggedness of each arch changes from low to high to low. The ruggedness of these arches varies, with the more rugged arches falling into sharper basins. This phenomenon is the same as the XOR dataset's. We can observe that residual MLPs have more rugged arches than non-residual MLPs, although the difference is not obvious in the Diabetes dataset.

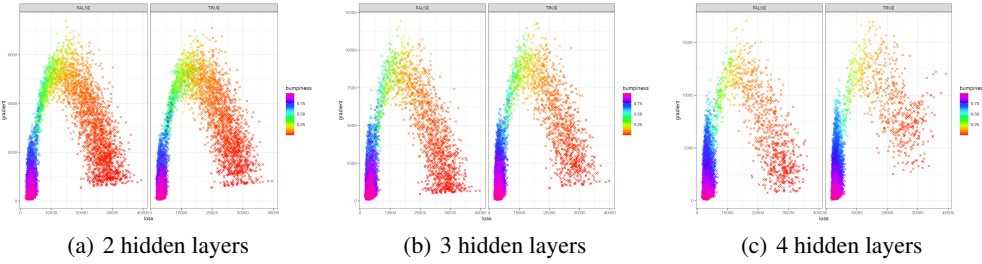

(a) 2 hidden layers      (b) 3 hidden layers      (c) 4 hidden layers

Figure 11: Diabetes, curvature change

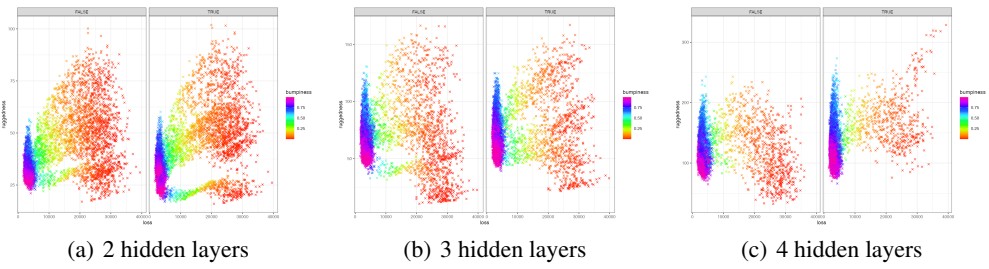

Figure 12: Diabetes, multi-layer structure of arches

Figure 13 reveals that as the depth of neural networks increases, so does the sharpness of the attraction basins. Residual MLPs have sharper basins than non-residual MLPs. Compared to the XOR and Iris datasets, the Diabetes dataset's loss landscapes are steeper and more rugged, with greater loss values.

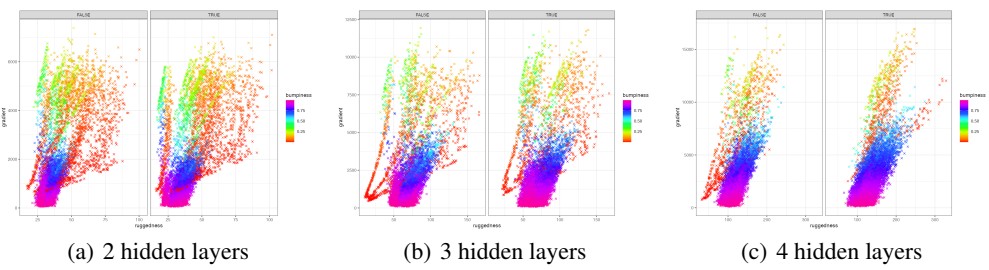

Figure 13: Diabetes, basins of attraction

## 6 CONCLUSIONS

Our work redefines the bumpiness and ruggedness metrics of points in the loss landscapes and figures out their correlation with the eigenvalues of the Hessian matrix of the loss function. We also empirically analyze the loss landscapes of residual and non-residual MLPs using the Hessian clouds based on the bumpiness and ruggedness metrics. The analysis indicates that as the network depth increases, the loss landscapes become steeper and more rugged, and this phenomenon is also found in the work of Bosman et al. (2023b). The macrostructures of the loss landscapes exhibited by various datasets vary substantially.

In addition, the effect of residual connections is studied in our work. Residual MLPs generate more and sharper lower-loss attraction basins in the XOR dataset, sharper basins in the Iris dataset, and sharper basins in the Diabetes dataset compared to non-residual MLPs. Additionally, residual MLPs tend to produce more rugged, steeper, and higher-loss landscapes than non-residual MLPs, but residual MLPs can converge to low-loss basins in the end. These phenomena may indicate that residual connections are advantageous for accelerating the convergence of training and reducing the eventual loss to which the training converges. The specifics should be analyzed based on the training datasets.

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
