# Appendix for Visual Analysis of the Bumpiness and Ruggedness of Residual Neural Network Landscapes

## A    Appendix

### A.1    loss-gradient clouds with bumpiness

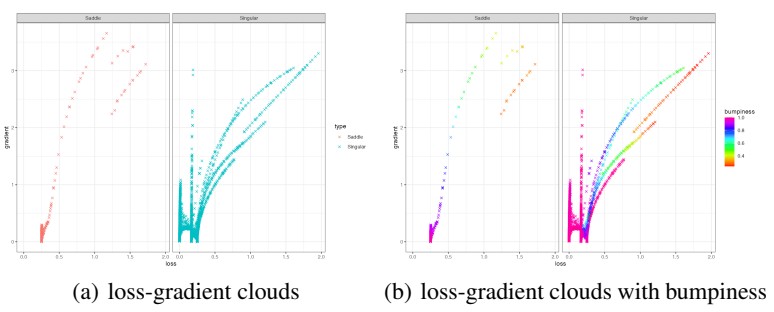

(a) loss-gradient clouds        (b) loss-gradient clouds with bumpiness

Figure 1: Figure (a) is the loss-gradient cloud. Figure (b) shows that as the optimization algorithm progresses, the curvature of the landscapes, which are predominantly singular, approaches convexity.

### A.2    Convex, concave, saddle, singular points in the XOR, Iris, Diabetes datasets

Table 1: Statistics of different types of sampling points

| Datasets | Number of hidden layers | Residual | Number of convex points | Number of concave points | Number of saddle points | Number of singular points |
|---|---|---|---|---|---|---|
| XOR | 2 | with | 0 | 0 | 103 | 9897 |
| | | without | 0 | 0 | 1014 | 8986 |
| | 3 | with | 0 | 0 | 0 | 10000 |
| | | without | 0 | 0 | 0 | 10000 |
| | 4 | with | 0 | 0 | 0 | 10000 |
| | | without | 0 | 0 | 0 | 10000 |
| Iris | 2 | with | 0 | 0 | 10 | 29990 |
| | | without | 0 | 0 | 957 | 29043 |
| | 3 | with | 0 | 0 | 0 | 30000 |
| | | without | 0 | 0 | 0 | 30000 |
| | 4 | with | 0 | 0 | 0 | 30000 |
| | | without | 0 | 0 | 0 | 30000 |
| Diabetes | 2 | with | 0 | 0 | 0 | 10000 |
| | | without | 0 | 0 | 0 | 10000 |
| | 3 | with | 0 | 0 | 0 | 10000 |
| | | without | 0 | 0 | 0 | 10000 |
| | 4 | with | 0 | 0 | 0 | 10000 |
| | | without | 0 | 0 | 0 | 10000 |

### A.3    Examples of Hessian Clouds

Below is a comprehensive illustration of the Hessian clouds of the MLPs with two hidden layers utilized in our experiments. To illustrate the trajectory of the sampling, we change the scatters' color to the corresponding step.

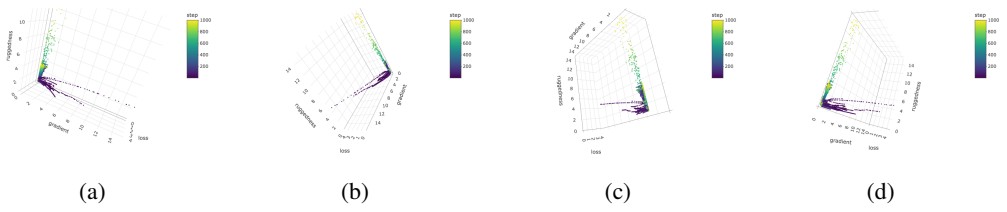

(a)      (b)      (c)      (d)

Figure 2: Hessian clouds, XOR, 2 hidden layers

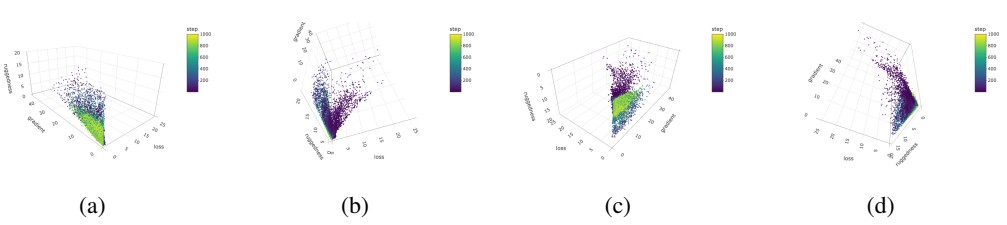

(a)      (b)      (c)      (d)

Figure 3: Hessian clouds, Iris, 2 hidden layers

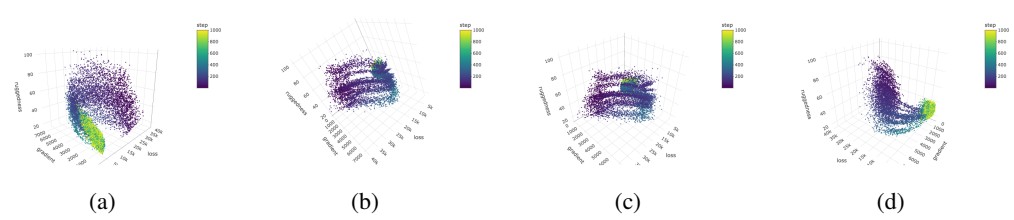

(a)      (b)      (c)      (d)

Figure 4: Hessian clouds, Diabetes, 2 hidden layers

### A.4 V-SHAPED OSCILLATION AREAS IN THE HESSIAN CLOUDS OF THE XOR DATASET

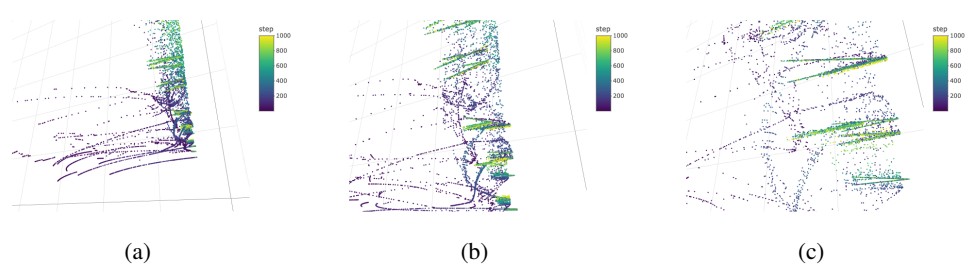

(a)      (b)      (c)

Figure 5: V-shaped oscillation areas, XOR, 2 hidden layers

## A.5  ANOTHER BASIN IN NON-RESIDUAL NN LANDSCAPES OF THE IRIS DATASET

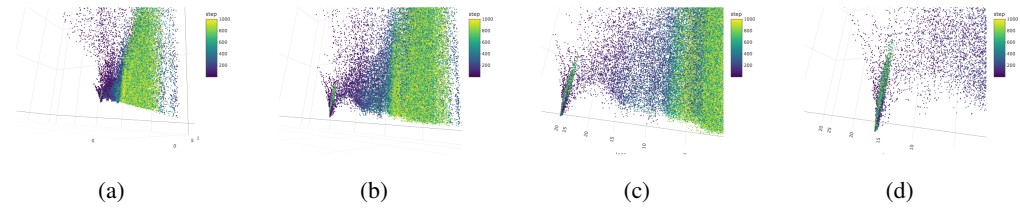

(a)      (b)      (c)      (d)

Figure 6: another basin, Iris, 2 hidden layers