# OpenReview forum: "Visual Analysis of the Bumpiness and Ruggedness of Residual Neural Network Landscapes"
_ICLR.cc/2024/Conference — ICLR 2024 Conference Withdrawn Submission_

### Official Review · Reviewer_PtGd · 2023-10-28

**Soundness:** 2 fair
**Presentation:** 2 fair
**Contribution:** 2 fair
**Rating:** 3
**Confidence:** 2

**Summary:**

The paper analyzes the effect of residual connections in MLPs on the loss landscape. The authors challenge the common belief that residual connections lead to smoother loss landscape and present an analysis showing the opposite. To this end, the authors introduce new notions of bumpiness and ruggedness and derive some properties. The paper further analyzes the evolution of these measures throughout training. Small MLPs with 2, 3, and 4 layers are analyzed on three low-dimensional data sets.

**Strengths:**

Better understanding the training dynamics and architectural design choices of modern neural network architectures is an important endeavor. Challenging common assumptions about relevance of certain design choices and procedures such as residual connections can lead to broad impact and improvements across subfields.

**Weaknesses:**

- The results are hard to follow. The axis labels and ticks in he figures are barely readable; side-by-side plots have differently scaled axes without giving a rationale for this, which makes it hard to draw conclusions from them. I assume “True”/”False” refers to networks with and without residual connections? If so, it would be good to make this explicit.
- Some of the design choices in the experiments are hard to understand for me. For example it is not discussed how using non-smooth ReLU activation functions affect their theory. To get around this, the authors could use e.g. the GELU activation function.
- Similarly, modern MLPs can have a latent dimension different from the input dimension (replacing the skip connection with a linear map to accommodate the change in width). I feel the latent dimension of 2, 4, and 10 might not be representative for modern MLPs used in practice.
- The authors use the optimization algorithm from (Bosman et al. 2018a). How does this differ from SGD or ADAM and other more popular algorithms? Why did the authors not use one of those?
- It is unclear from the results presented in the paper whether the presented networks reach reasonable accuracy on the corresponding tasks.

**Questions:**

If I understand correctly, the proposed definition of bumpiness defined in Sec. 3.1 removes the absolute values to make it sensitive to whether the function is locally convex or concave. I’m not convinced that this definition always leads to this outcome, it would be great if the authors could give some additional intuition.

---

### Official Review · Reviewer_TMom · 2023-10-29

**Soundness:** 2 fair
**Presentation:** 1 poor
**Contribution:** 2 fair
**Rating:** 3
**Confidence:** 4

**Summary:**

This paper proposes a new type of visualization of Hessian clouds, as well as derive the bumpiness and ruggedness metrics. By using the proposed visualization, it empirical compare the loss landscape of MLPs with or without residual connection.

**Strengths:**

* The motivation of the paper is to study the loss landscape via visualization meets general interests.

**Weaknesses:**

* It doesn't seem to be a paper that is ready for a machine learning / deep learning conference. It focuses on the visualization and there is no clear conclusion or insights from the results.
* The visualization figures to follow and read. It would be very helpful if the authors could add more detailed explanations in each caption to show how the displayed patterns related to the insights and conclusions.
* There is no model performance-related metric (e.g. accuracy) shown in the paper, which makes it hard to know if the proposed metrics and visualization are really meaningful for practical usage.

**Questions:**

* What is the motivation for proposing the new sharpness metrics bumpiness and ruggedness, what is the advantage of it compared to the conventional metrics like Hessian trace/top-eigenvalue?
* What is the computation complexity of the proposed visualization? Is it possible to scale up to larger models?

**Details Of Ethics Concerns:**

No ethical concern.

---

### Official Review · Reviewer_Me3L · 2023-10-29

**Soundness:** 3 good
**Presentation:** 2 fair
**Contribution:** 2 fair
**Rating:** 3
**Confidence:** 5

**Summary:**

The authors extend an existing loss landscape visualisation technique (loss-gradient clouds) to incorporate ruggedness and bumpiness metrics, and study the effects of residual connections on MLPs using their proposed visualisation (Hessian clouds). The main contribution of the paper is the proposition of two new metrics to quantify landscape properties, as well as the extension of the existing visualisation method. Observations regarding the residual connections seem to mostly corroborate with existing insight.

**Strengths:**

The paper is focused on enhancing existing visualisation methods, which is great: our tools are limited in the visualisation domain, and the topic is well worth the exploration. The metrics proposed by the authors (ruggedness and bumpiness) are mathematically sound. Visualisations are quite striking, illustrating that scatter plots applied to scalar representations of the NN loss landscapes can be a very effective tool to visually study these extremely high-dimensional problems.

**Originality:** the proposed metrics for ruggedness and bumpiness are original, and clearly provide a new angle of view on the loss landscape. The visualisation itself (Hessian cloud) is a derivative of the loss-gradient cloud, and the residual connections seem to exhibit expected behaviour.

**Quality and clarity:** the paper is mostly well-written, although clarity can be improved. See below my comments on explicitly referencing the appendices.

**Significance:** in my opinion, authors could have focused on something more enigmatic than the residual connections - for example, the effect of regularisation methods such as dropout on the landscape. The proposed metrics are new and interesting, but the subject matter (residual connections) does not offer that much new insight.

**Weaknesses:**

Related work is a bit haphazard: the authors cite numerous papers, but do not structure the review into a coherent narrative.

Authors refer to various appendices (Appendix A.1, Appendix A.2, etc.) but I could not find either of the listed appendices in the paper. I realise that the appendices are included as supplementary material, but explicit references to supplementary material simply do not work. Can the authors perhaps add appendices directly to the paper, otherwise refrain from discussing appendix data explicitly? The paper must be self-contained, and in its current form the paper cannot be followed without the supplementary material.

Authors propose measures for bumpiness and ruggedness, but do not provide a clear intuition for the difference between the two. Adding a toy 1D example just to illustrate the difference between “bumpy” and “rugged” can be very beneficial for the narrative.

Author’s choice of study subject (MLP with residual connections) is not that interesting: at the end of the day, the paper simply shows that residual connections make gradients steeper (expected). Authors also state that residual connections increase the ruggedness in MLPs, which was shown to not be the case for CNNs, which is an interesting observation, if true. In my opinion, one of the plots is not interpreted correctly in this regard. Either way, authors conclude that residual connections make training easier, which is once again a known fact. Thus, although a new lens to study loss landscapes is offered, the lens falls short of discovering something of importance.

Authors only make use of their proposed ruggedness/bumpiness metrics, which seems short-sighted. Ruggedness can be measured in terms of the Hessian trace or entropy of the walk - how do these existing metrics correlate with the proposed metric? A visual and numerical comparison would have enhanced the paper significantly. There is only one reason for new metrics: if the old metrics fail.

**Questions:**

Authors state that ruggedness is “an essential metric in measuring the network performance”. Is this true? Could you provide evidence that ruggedness is strongly correlated with network performance?

Figure 3: What do the “TRUE” and “FALSE” labels in the figure represent? Please discuss explicitly. Authors state that Figure 3 shows that residual connections yield more rugged landscapes - if the right pane corresponds to residual connections, then I disagree with the interpretation. It seems that residual connections yield stronger gradients (expected and desired effect), as well as overall lower ruggedness except for a few outliers.

What is the difference between Figures 4, 5, and 6? All the axis (x,y,z) are identical, yet the observed patterns are different. How was the data generated for each of the plots?

Abstract: “beneficial to generating” -> beneficial for generating

Abstract: “residual MLPs prefer to generate” -> residual MLPs generate (MLPs do not have preferences, since they do not have agency)

“people’s understanding” - the understanding

“and it is a two-dimensional visualization” - which is a two-dimensional visualization

“the two-dimensional presentation method can capture some characteristics of the loss landscapes, but the results represented are still not clear enough” - what do you mean by “not clear enough”? This is a very vague statement.

“as discussed in Introduction 1” - as discussed in Section 1.